# Impact of Soil Surface Temperature on Changes in the Groundwater Level

**Mukhamadkhan Khamidov** [1,†], **Javlonbek Ishchanov** [1,*,†], **Ahmad Hamidov** [1,2], **Ermat Shermatov** [3] and **Zafar Gafurov** [4]

1  Department of Irrigation and Melioration, "Tashkent Institute of Irrigation and Agricultural Mechanization Engineers" National Research University, Tashkent 100000, Uzbekistan; khamidov_m@mail.ru (M.K.); ahmad.hamidov@zalf.de (A.H.)

2  Research Area 3 "Agricultural Landscape Systems", Leibniz Centre for Agricultural Landscape Research, Eberswalder Straße 84, 15374 Müncheberg, Germany

3  Research Institute of Irrigation and Water Problems, Tashkent 100084, Uzbekistan; jkkchiv@mail.ru

4  International Water Management Institute—Central Asia Office, Tashkent 100000, Uzbekistan; z.gafurov@cgiar.org

*  Correspondence: javlon@email.com

†  These authors contributed equally to this work.

**Abstract:** The relationship between the soil surface temperature and groundwater level is complex and influenced by various factors. As the soil surface temperature increases, water evaporates quickly from the soil, which can lead to a decrease in the groundwater level. In this study, we analyzed the impact of soil surface temperature on changes in the groundwater level in the Bukhara region of Uzbekistan using data from 1991 to 2020. The Bukhara region experiences regular water shortages, increased soil salinization, and inefficient energy in lift-irrigated areas, which is a typical constellation of challenges to the water–energy–food–environment (WEFE) nexus. The soil surface temperature data were collected from the Hydrometeorological Service Agency, whereas groundwater level data were obtained from the database of the Amelioration Expedition under the Amu-Bukhara Basin Irrigation Systems Authority. We used linear regression analysis and Analysis of Variance (ANOVA) tests to establish the significance of the relationship between the soil surface temperature and groundwater level, as well as the impact of the location of the groundwater level measurements. The results indicate that the model was a good fit to the data, and both the intercept and the soil surface temperature were significant factors that affected groundwater level. The results further suggest that the strength of the relationship between solar radiation and soil surface temperature is very high, with a correlation coefficient of 0.840. This means that when solar radiation increases, soil surface temperature also tends to increase. The analysis also showed that 53.5% of the changes in groundwater level were observed by the regression model, indicating a moderately correlated relationship between the groundwater level and soil surface temperature. Finally, higher solar radiation leads to higher soil surface temperature and higher evapotranspiration rates, which can lead to a decrease in groundwater level. As a result, we observe that the soil surface temperature determines changes in the groundwater level in the study region.

**Keywords:** soil surface temperature; groundwater level; linear regression; correlation; ANOVA; water–energy–food–environment nexus; Uzbekistan

## 1. Introduction

The soil surface temperature is the temperature of the top layer of soil. It is influenced by a number of factors, including the air temperature, the amount of sunlight, and the wind speed [1]. Soil surface temperature can be measured through in situ ground investigations, as well as via satellite data [2,3]. Many scholars point to the fact that soil temperature at any depth beneath the earth's surface remains constant throughout the year [4,5]. However,

some studies indicate that soil temperatures at shallow depths present notable fluctuations on a daily and annual basis [6,7].

The amount of moisture in the soil, the temperature of the soil, and the temperature at the surface are important factors in hydrology that determine different land surface activities [8]. Soil moisture, land surface temperature, and their relationship can be influenced by multiple factors such as soil emissivity, evapotranspiration, and thermal inertia [9]. These parameters change over space and time, and variations in surface temperature can help to identify target properties [10]. Additionally, as soil temperature increases, the rate of evaporation also increases. This can lead to a decrease in soil moisture, especially in the surface layers of the soil. The soil temperature is inversely proportional to the soil moisture [11]. The removal of canopy cover and loss of organic matter can result in a reduction in soil moisture and an increase in soil temperature at the soil's surface [12].

The groundwater level plays a significant role in fulfilling the water requirements of crops, especially in arid and semi-arid zones [13,14]. The level of groundwater in an aquifer can be affected by a number of factors, including irrigation, which can add water to the ground and raise the water table; filtration, which can remove water from the ground and lower the water table; soil surface temperature, which can affect the rate of evaporation and thus the amount of water that is available to recharge the aquifer; and leakage of water into the drains, which can remove water from the ground and lower the water table [15]. The use of mathematical modeling can help us to obtain more precise results about the effects of climate change on water table depth and salinity changes in different regions [16,17]. Drying climate, intensive pumping for irrigation purposes, and changes in land use can impact the water table [18–20]. Climate change has caused longer and more intense heat waves, which reduce groundwater levels [21]. The relationship between the soil surface temperature and groundwater level is complex. In general, as the soil surface temperature increases, water evaporates more quickly from the soil. This can lead to a decrease in groundwater level [22,23].

Several variables can influence the relationship between soil surface temperature and groundwater level. These include the following: the soil type—sandy soils have a reduced ability to retain water compared with clay soils, making them more prone to evaporation; the amount of vegetation cover—vegetation helps to shade the soil and reduce evaporation; the amount of water in the soil—the more water that is in the soil, the less water will evaporate; the wind speed—wind can increase evaporation by blowing away water vapor; the amount of sunlight—sunlight provides the energy for evaporation; and the air temperature—the higher the air temperature, the more water will evaporate [24].

The impacts of the relationship between soil surface temperature and groundwater level can be significant and can have a number of negative ramifications for ecosystems and human society [25]. These impacts include the following:

- Reduced water availability—as soil surface temperature increases, water evaporates more quickly from the soil [26,27]. This can lead to a decrease in groundwater level. This can reduce the amount of water available for drinking, irrigation, and other uses;
- Increased risk of waterborne diseases—as water levels decrease, water becomes more stagnant and can become contaminated with bacteria and other microorganisms [28]. This can increase the risk of waterborne diseases such as cholera, typhoid fever, and diarrhea.

In this study, we investigated solar radiation as a factor contributing to the potential lifting of soil surface temperatures. This involved an assessment of the relationship between solar radiation and the soil surface temperature. Solar radiation is the energy that comes from the sun and reaches the earth's surface. It is the primary source of heat for the earth's surface and atmosphere. When solar radiation hits the earth's surface, it is absorbed by the soil, rocks, and other objects. This absorption of solar radiation causes the earth's surface to heat up [29].

The relationship between solar radiation and soil surface temperature is a strong positive one. This means that as solar radiation increases, so does the soil surface tem-

perature [30]. This relationship is influenced by a number of factors, including the time of the day, the season, the latitude, the presence of clouds, the air temperature, and the soil moisture. Our study focuses on the relationship between soil surface temperature and groundwater measurements because solar radiation does not directly impact to the changes on groundwater levels.

The current research aims to examine the relationship between soil surface temperature and groundwater level through analyzing data collected throughout the growing season, spanning from April to September, between the years 1991 and 2020. The specific objectives of this research are (1) to comprehend the changes in groundwater level and soil surface temperature over time; and (2) to establish a statistical correlation between soil surface temperature and groundwater level using data collected from the Hydrometeorological Service Agency.

The main hypothesis of this research is that soil surface temperature has a significant impact on changes in groundwater level in the study area.

## 2. Materials and Methods

### 2.1. Case Study Area

The Bukhara region was selected as a study area for this research; it is located in the southwest of Uzbekistan (Figure 1). The Kyzyl Kum desert occupies a significant portion of the land and shares borders with Turkmenistan, the Navoiy region, the Kashkadarya region, a small section of the Khorezm region, and the Karalpakstan Republic. It spans a vast area measuring 40,216 km$^2$ [31]. The current population is approximately 1.97 million (as of 2022), with around 63% residing in rural regions [32].

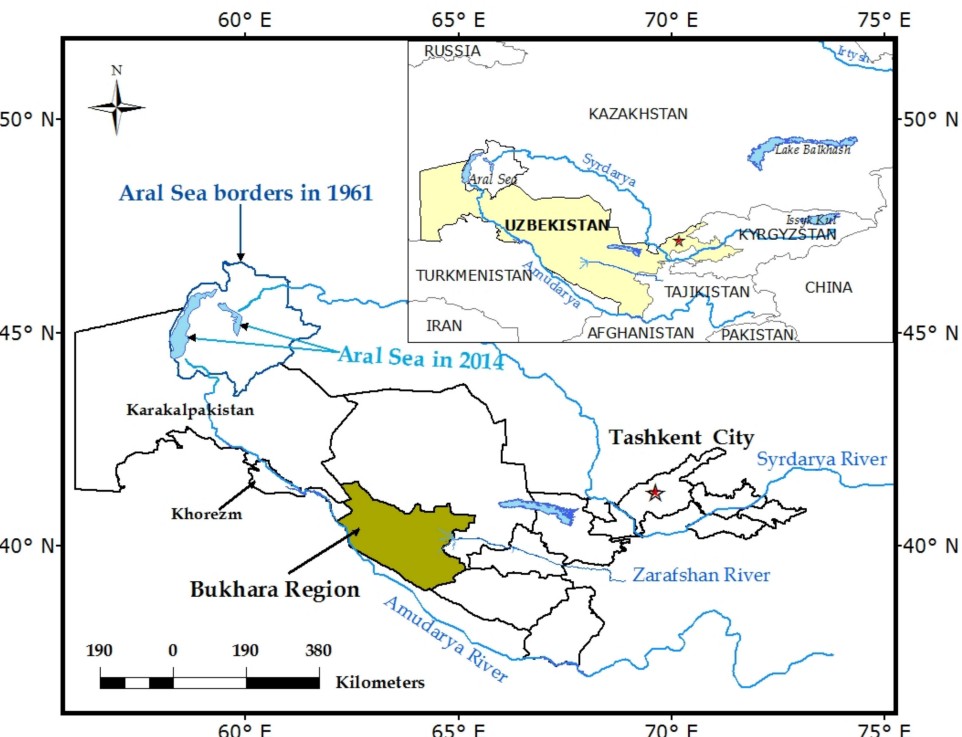

**Figure 1.** Location of the study area.

The region is composed of 11 administrative districts and 2 cities at the district level. The city of Bukhara serves as the capital, with an estimated population of 284,100 in 2021. Other significant towns in the region include Olot, Karakul, Bukhara, Jondor, Gijduvon, Kagan, Romitan, Karavulbazar, Shofirkon, and Vobkent. The climate in Bukhara is typically characterized as arid and continental [33]. The region has more than 230,000 ha of irrigated land, consuming over 5 km$^3$ of Amudarya river water [34].

Since the region is located in an arid zone, with high soil salinity and temperature, for the first time, we here investigated the effects of soil surface temperature on groundwater level.

In terms of soil characteristics, soils in the region are mostly clay, medium and heavy loamy soils—56.9%, light loamy soils—35.6%, and sand and loamy soils—7.5%. Soils in the Bukhara region are characterized by a very low humus content (1–2%) [35]. Overall, Bukhara has about 11 different major soil types, which can be sorted into four different classifications [36]:

(i)    Sandy and desert sandy soils—Most soils in the western and southern parts of the Bukhara region (including in the Kyzyl-Kum desert and the Uzbek–Turkmen border area) have low levels of humus (0.5%) and nitrogen (0.04–0.05%);

(ii)   Grey-brown soils—The most common soil types in the northern and eastern parts of the region are grey-brown soils, which are used for intensive irrigation farming under arid climates. These soils have a variety of textures, from sandy–loamy to medium loamy. The humus content in the top layer of soil varies in the range 0.6–0.9%, but it is higher in traditionally irrigated areas (1.2–1.8%). The nitrogen content in these soils is between 0.05% and 0.16%, and the total phosphorus content is between 0.09% and 0.11% [35];

(iii)  Takyr soils—These rare soil types are found mostly in the central Bukhara region, mixed with grey-brown and sandy soils. They form in shallow, clay-rich depressions that collect water. As the water evaporates, salt crusts form on the surface;

(iv)   Meadow soils—Meadow soils are found along the Zarafshan River and its former riverbed, extending to the Amu Darya River. They have higher humus (1.1–1.4%) and nitrogen (0.08–0.12%) contents than other types of desert soils.

Agriculture is the predominant consumer of water resources, accounting for the majority of water usage. In contrast, industrial and municipal water use combined make up less than 10% of the overall water consumption. The Amu-Bukhara Basin Irrigation Systems Authority (BISA) is the main state water organization that manages and supplies irrigation water to the entire region.

In this research, the data were analyzed during the growing season from April to September of each year during 1991–2020. The soil surface temperature, air temperature, precipitation, and solar radiation data were collected from the Hydrometeorological Service Agency of the Republic of Uzbekistan. Groundwater level data were taken from the database of the Amelioration Expedition under the Amu-Bukhara BISA. Groundwater level data were obtained from 2590 monitoring wells within the Bukhara region using monthly average values from 1991 to 2020 (Table 1).

**Table 1.** Groundwater monitoring wells located in the Bukhara region.

| No. | Administrative Districts | Number of Monitoring Wells |
|---|---|---|
| 1 | Vobkent | 244 |
| 2 | Gijduvon | 326 |
| 3 | Jondor | 303 |
| 4 | Kagan | 130 |
| 5 | Karavulbazar | 113 |
| 6 | Karakul | 222 |
| 7 | Olot | 234 |
| 8 | Romitan | 263 |
| 9 | Shofirkon | 268 |
| 10 | Bukhara | 267 |
| 11 | Peshku | 220 |
| | Total for Bukhara region | 2590 |

Monitoring wells are used to measure the level and salinity of groundwater. They are important for understanding how groundwater is being used and managed, and for

detecting and responding to changes in groundwater levels or salinity. The high number of monitoring wells in the Bukhara region reflects the importance of groundwater to agriculture in the region. The region is a major agricultural zone of Uzbekistan, and groundwater is a major source of irrigation water.

Table 1 shows that the number of monitoring wells varies from district to district. The districts with the most monitoring wells are Vobkent, Gijduvon, Jondor, and Shofirkon. These districts are all located in the center of the Bukhara region, and they are all major agricultural areas. The districts with the fewest monitoring wells are Kagan, Karavulbazar, and Peshku. These districts are located on the periphery of the Bukhara region, and they have a lower level of agricultural development.

*2.2. Linear Regression Model*

In this study, we used a linear regression model to analyze the connection between soil surface temperature and groundwater level changes because it is a simple and effective method for modeling linear relationships between variables. Linear regression models assume that the relationship between the variables is linear, meaning that it can be represented by a straight line. Moreover, linear regression is a simple and easy-to-understand statistical model. It is also relatively easy to interpret the results of a linear regression analysis. Also, the slope coefficient (b) in the linear regression equation represents the change in groundwater level for every unit change in soil surface temperature. This is a very interpretable parameter, which makes linear regression a good choice for modeling the relationship between these two variables. The relationship between variables in a biological object can be depicted and analyzed by using symbols such as "x" and "y" and representing them as points on a coordinate system. This allows for the creation of a scatter diagram, which enables the assessment of the shape and strength of the connection between different attributes. In essence, this approach facilitates the evaluation of correlations in biological data [37].

In most instances, this association tends to form a straight line or can be approximated as such. The relationship between the variables x and y is represented by a linear equation in the form of $y = a + bx_1 + cx_2 + dx_3 + \ldots$, where a, b, c, d... determine how the equation's parameters, $x_1$, $x_2$, $x_3$..., relate to the y-function [38].

The connection between groundwater level (y) and soil surface temperature (x) can be depicted using a basic linear regression model:

$$y = a + bx \tag{1}$$

In the equation for the linear regression, the b-parameter is responsible for determining the slope of the regression line in relation to the coordinate axes, which can be thought of as a shift along the a-coordinate axis. In the field of analytical geometry, this parameter is commonly referred to as the angle coefficient, while in biometrics, it is known as the regression coefficient.

## 3. Results and Discussion

The study's findings for average annual air temperature indicate that the minimum mean annual air temperature was 14.6 °C, the maximum was 17.1 °C, and the mean was 16.1 °C during 1991–2020. The data for air temperature were collected from the Bukhara Meteorological Station. The standard deviation was 0.563 °C (Table 2).

**Table 2.** Analysis of annual mean air temperature from 1991 to 2020.

| Variable | Minimum | Maximum | Mean | Std. Deviation |
|---|---|---|---|---|
| Mean annual air temperature | 14.675 | 17.175 | 16.084 | 0.563 |

Figure 2 shows the change in annual average air temperature in the Bukhara region from 1991 to 2020. The average air temperature in Bukhara has been increasing over the

past 20 years, with a particularly sharp increase in the last decade. In 1991, the average air temperature in Bukhara was 15.3 °C. By 2020, the average air temperature had increased to 17.2 °C. This represents an increase of over 1.9 °C in just 30 years.

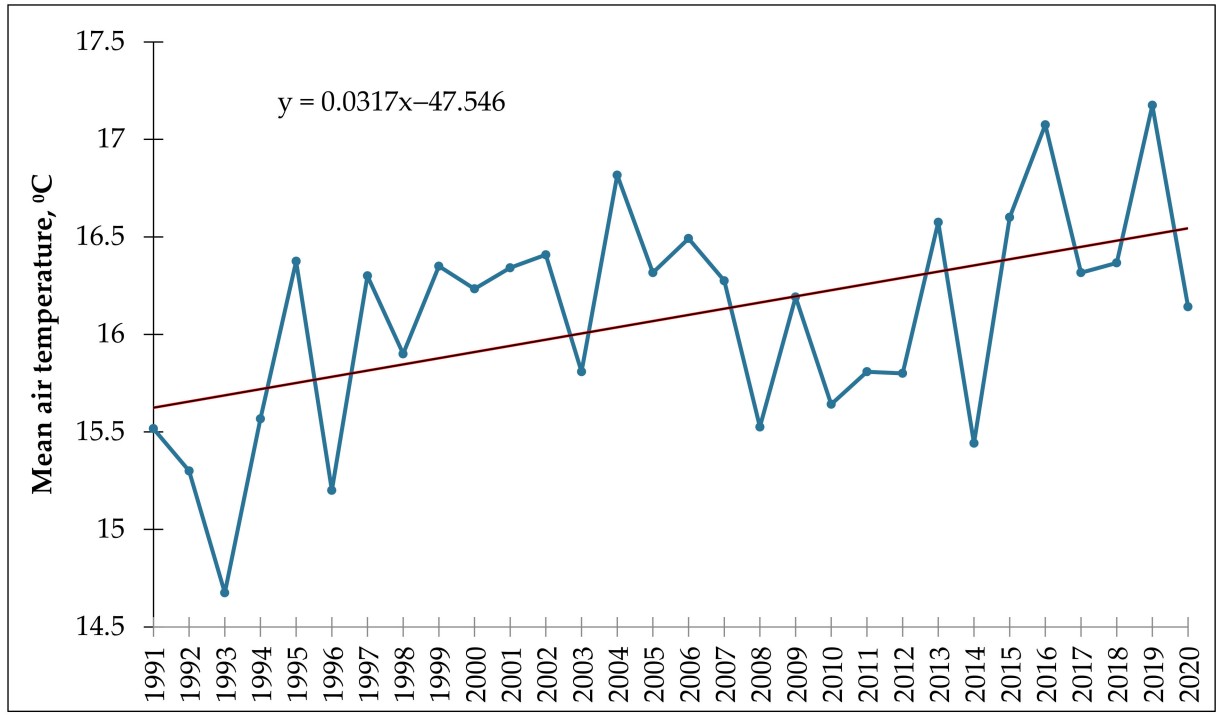

**Figure 2.** Change of annual average air temperature in Bukhara region from 1991 to 2020.

Furthermore, the annual and growing season rainfall values were analyzed for the period of 1991–2020. The annual precipitation is the total amount of precipitation that falls in a year, while the growing season precipitation is the amount of precipitation that falls during the vegetation season, which is typically defined as the period from April to September (Figure 3). The figure shows that the annual precipitation in Bukhara has varied significantly over the past 30 years, from a minimum of 60.5 mm in 2008 to a maximum of 207.9 mm in 2020. The growing season precipitation has also varied significantly, from a minimum of 1.0 mm in 2001 to a maximum of 124.9 mm in 2020. The mean annual precipitation in the region over the past 30 years was 127.58 mm, and the mean growing season precipitation was 35.74 mm. The standard deviation of the annual precipitation was 39.12 mm, and the standard deviation of the growing season precipitation was 29.85 mm. This figure provides a useful overview of the annual and growing season precipitation in Bukhara region over the past 30 years. It can be used to assess the variability of precipitation in the region and to identify trends over time. This information can be used to inform decisions about water management and agricultural practices.

In addition, changes in the level of groundwater during the annual and vegetation period from 1991 to 2020 were studied. The annual groundwater level is the average depth of the water table over the entire year, while the growing season groundwater level is the average depth of the water table during the six months of the growing season (Figure 4).

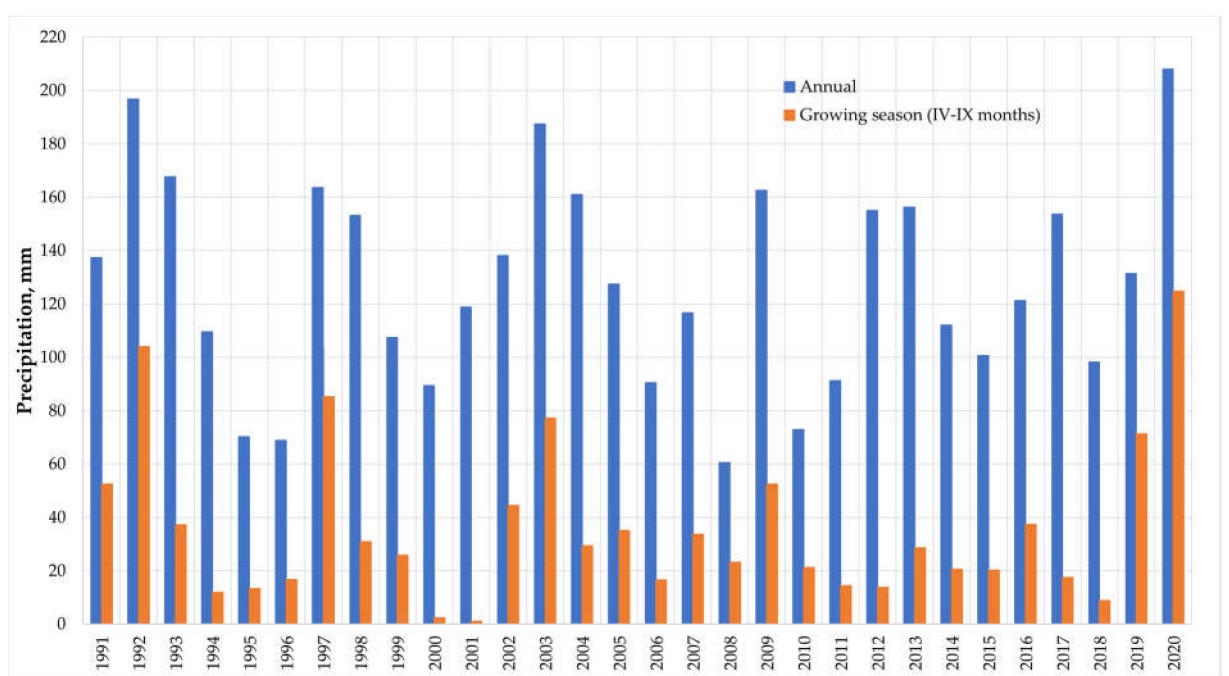

**Figure 3.** Changes in annual and growing season precipitation in Bukhara during 1991–2020.

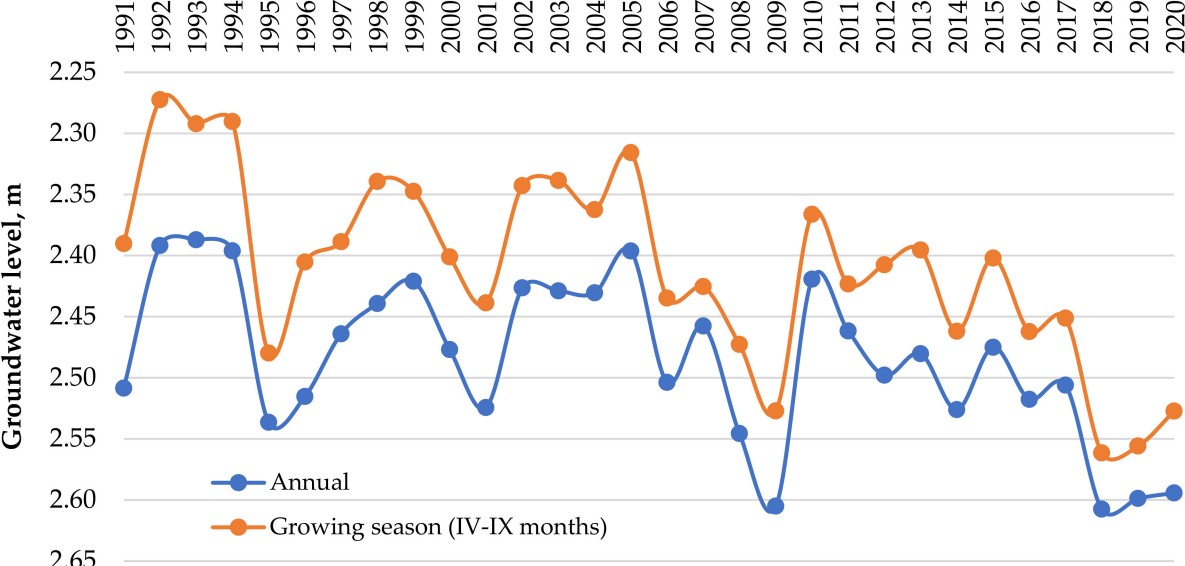

**Figure 4.** Changes in groundwater level during the annual and growing season in Bukhara from 1991 to 2020.

Figure 4 shows that the annual groundwater level in the region has decreased over the past three decades. In 1991, the annual groundwater level was around 2.5 m below the surface. By 2020, it had decreased to around 2.6 m below the surface. The growing season groundwater level has also decreased over time. In 1991, the growing season groundwater level was around 2.4 m below the surface. By 2020, it had decreased to around 2.55 m below the surface. Additionally, the decline in groundwater level has been more pronounced during growing seasons than during the rest of the year. This is likely due to the increased demand for water for irrigation during the growing season. Overall, the decline in groundwater level in Bukhara is a serious concern, as it could have a negative impact on agricultural production and water supply for human consumption. A decrease in the groundwater level can be associated with the intensive use of advanced water-saving

irrigation technologies (drip and sprinkler irrigation) and increased risks of water scarcity and climate change.

Furthermore, given that solar radiation does not have a direct impact on groundwater, we conducted a study concerning soil surface temperature. Table 3 shows a correlation matrix that illustrates the correlation coefficients between two variables: solar radiation and soil surface temperature.

**Table 3.** Correlation matrix.

| Variable | Solar Radiation | Soil Surface Temperature |
|---|---|---|
| Solar radiation | 1 | 0.840 |
| Soil surface temperature | 0.840 | 1 |

The correlation coefficient is a measure of the strength of the relationship between two variables. A correlation coefficient of 1 indicates a perfect positive correlation, meaning that the two variables always move in the same direction. In Table 3, the correlation coefficient between solar radiation and soil surface temperature is 0.840. This indicates a strong positive correlation between the two variables. In other words, solar radiation and soil surface temperature tend to move in the same direction. When solar radiation increases, soil surface temperature also tends to increase. And when solar radiation decreases, soil surface temperature also tends to decrease. This is because solar radiation is the main source of energy for heating the earth's surface. When more solar radiation reaches the earth's surface, the ground absorbs more heat and warms up. When less solar radiation reaches the earth's surface, the ground absorbs less heat and cools down.

However, it is important to note that correlation does not equal causation. Just because two variables are correlated does not mean that one causes the other. There could be other factors that are causing the changes in both variables. For example, changes in weather patterns could be causing changes in both solar radiation and soil surface temperature.

To determine whether there is a causal relationship between solar radiation and soil surface temperature, further research would be needed. This research could involve conducting experiments to isolate the effects of solar radiation on soil surface temperature.

Table 4 displays the results of a regression analysis of the variable "soil surface temperature".

**Table 4.** Regression of variable "soil surface temperature".

| Goodness of Fit Statistics (Soil Surface Temperature, °C) | |
|---|---|
| Observations | 11 |
| Sum of weights | 11 |
| DF (Degrees of freedom) | 9 |
| $R^2$ (Regression) | 0.706 |
| Adjusted $R^2$ | 0.674 |
| MSE (Mean squared error) | 0.117 |
| RMSE (Root mean squared error) | 0.343 |
| MAPE (Mean absolute percentage error) | 1.528 |
| DW (Durbin–Watson) [39] | 1.482 |
| Cp (Mallows's Cp coefficient) [40] | 2.000 |
| AIC (Akaike information criterion) [41] | −21.777 |
| SBC (Schwarz Bayesian criterion) [42] | −20.981 |
| PC (Partial correlation) | 0.424 |
| Press (Predicted residual error sum of squares) | 1.765 |
| $Q^2$ (Second quartile) | 0.509 |

The regression of the variable soil surface temperature has a good fit, with an $R^2$ value of 0.706 (Figure 5). This means that 70.6% of the variation in soil surface temperature can be explained by the regression model. The adjusted $R^2$ value is 0.674, which means that 67.4%

of the variation in soil surface temperature can be explained by the model after taking into account the number of independent variables in the model.

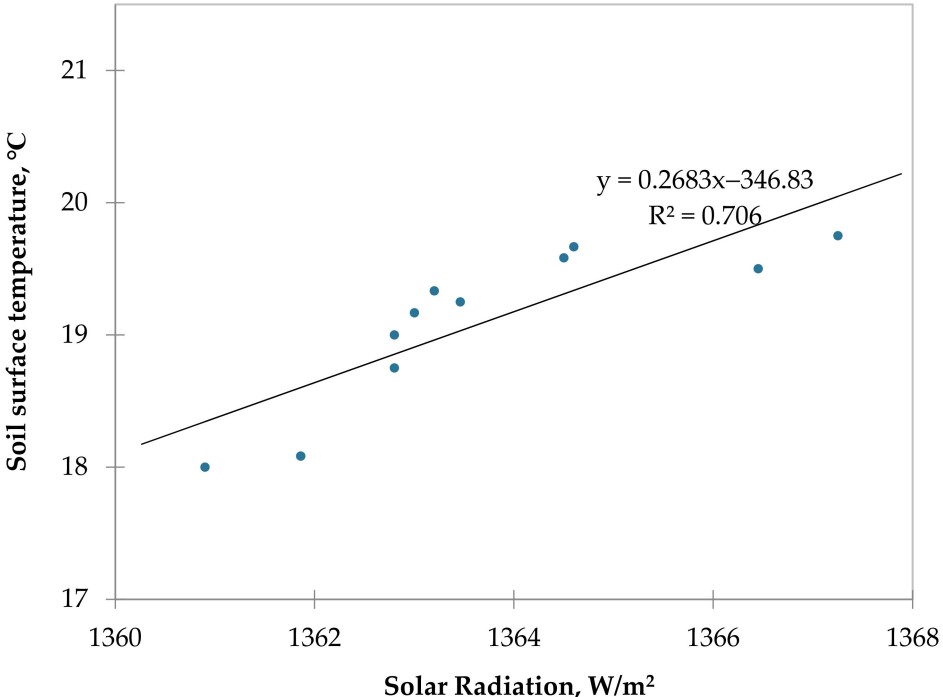

**Figure 5.** The relationship between solar radiation and soil surface temperature in the Bukhara region of Uzbekistan.

The mean squared error (MSE) is 0.117, which means that the average squared difference between the predicted and actual soil surface temperatures is 0.117. The root mean squared error (RMSE), a measure of the standard deviation of the residuals, is 0.343. The mean absolute percentage error (MAPE) is 1.528, which means that on average, the predicted soil surface temperatures are 15.28% different from the actual values.

The Durbin–Watson statistic (DW) is 1.482, which is close to 2.0, indicating that there is no autocorrelation between the residuals. The Akaike information criterion (AIC), a measure of how well the model fits the data, is −21.777. A lower AIC value indicates a better fit. The Schwarz Bayesian criterion (SBC), another measure of how well the model fits the data, is −20.981. A lower SBC value indicates a better fit.

The PRESS statistic, a measure of how well the model predicts new data, is 1.765. A lower PRESS value indicates a better prediction. The $Q^2$ statistic, a measure of how well the model explains the variance in the dependent variable, is 0.509. A higher $Q^2$ value indicates a better explanation.

Overall, the regression of the variable soil surface temperature has a good fit and is able to explain a significant amount of the variation in soil surface temperature. The model can be used to predict soil surface temperatures with a reasonable degree of accuracy.

Table 5 shows the standardized coefficients for the relationship between solar radiation and soil surface temperature.

**Table 5.** Standardized coefficients (soil surface temperature).

| Variable | Value | Standard Error | t | Pr > \|t\| | Lower Bound (95%) | Upper Bound (95%) |
|---|---|---|---|---|---|---|
| Solar radiation | 0.840 | 0.181 | 4.651 | 0.001 | 0.432 | 1.249 |

The standardized coefficient is a measure of the strength of the relationship between two variables, after taking into account the standard deviations of both variables.

In Table 5, the standardized coefficient for solar radiation is 0.840. This means that for every 1 standard deviation increase in solar radiation, there is a 0.840 standard deviation increase in soil surface temperature.

The standard error of the coefficient is 0.181. This means that we can be 95% confident that the true standardized coefficient is within 0.181 of 0.840.

The t-statistic is 4.651. This is greater than the critical t-value of 2.776, so we can reject the null hypothesis that the true standardized coefficient is equal to 0. In other words, there is a statistically significant relationship between solar radiation and soil surface temperature.

The *p*-value is 0.001, which is less than 0.05. This means that the probability of obtaining the observed standardized coefficient by chance is less than 0.05. Therefore, we can conclude that the relationship between solar radiation and soil surface temperature is statistically significant.

The lower bound of the 95% confidence interval is 0.432 and the upper bound is 1.249. This means that we can be 95% confident that the true standardized coefficient is between 0.432 and 1.249.

In conclusion, Table 4 shows that there is a statistically significant positive relationship between solar radiation and soil surface temperature. This means that an increase in solar radiation is associated with an increase in soil surface temperature.

Below, we will consider the relationship between the mean soil surface temperature and the level of groundwater in the months of April–September in Bukhara in 1991–2020. We derived 117 observations for each variable. The minimum groundwater level was 1.720 m, and the maximum was 2.29 m. The mean groundwater level was 2.051 m, and the standard deviation was 0.132 m. The minimum soil surface temperature was 18.0 °C, and the maximum was 39.0 °C. The mean soil surface temperature was 30.444 °C, and the standard deviation was 4.996 °C (Table 6).

**Table 6.** Summary statistics.

| Variable | Observations | Minimum | Maximum | Mean | Std. Deviation |
|---|---|---|---|---|---|
| Groundwater level, m | 117 | 1.720 | 2.29 | 2.051 | 0.132 |
| Soil surface temperature, °C | 117 | 18.0 | 39.0 | 30.444 | 4.996 |

The data in Table 6 suggest that the groundwater level and soil surface temperature varied significantly over the time period studied. The groundwater level was highest in the middle of the growing season (July) and lowest at the beginning and end of the growing season (April and September). The soil surface temperature was highest in the middle of the day (noon) and lowest at night.

The data in Table 6 can be used to track changes in groundwater level and soil surface temperature over time. This information can be used to understand the effects of climate change on these important environmental variables.

*ANOVA Test*

Table 7 provided below is an ANOVA table. ANOVA is a statistical test that is used to compare the means of two or more groups. It has been used for global sensitivity analyses for various models [43]. It can be used to measure the statistical significance of the difference between two different methods or more [44]. In this case, the ANOVA is being used to compare the groundwater levels of different locations.

Table 7 has seven columns:

- Source—This column identifies the source of variation in the data. In this case, the source of variation is the location of the groundwater level measurements;

- DF—This column stands for degrees of freedom. Degrees of freedom are a measure of the variability in the data. In this case, there are 1.0 degrees of freedom for the model and 115.0 degrees of freedom for the error;
- Sum of squares—This column shows the amount of variation in the data that is explained by the model. In this case, the model explains 1.077 of the variation in the data;
- Mean squares—This column is calculated by dividing the sum of squares by the degrees of freedom. In this case, the mean square for the model is 1.077;
- F—This column is the F statistic. The F statistic is a measure of the significance of the model. In this case, the F statistic is 132.506;
- Pr > F—This column is the *p*-value. The *p*-value is a measure of the probability of obtaining the observed results if the null hypothesis is true. In this case, the *p*-value is <0.0001;
- Signification codes—This column shows the significance of the results. A significance code of * indicates that the results are significant at the 0.05 level. A significance code of ** indicates that the results are significant at the 0.01 level, and a significance code of *** indicates that the results are significant at the 0.001 level.

In this case, the *p*-value is <0.0001, which is less than 0.05. This means that the results are significant at the 0.05 level. Therefore, we can conclude that the model is significant and that the location of the groundwater level measurements does have an impact on the groundwater level.

**Table 7.** Analysis of variance (groundwater level, m).

| Source | DF | Sum of Squares | Mean Squares | F | Pr > F | *p*-Value Signification Codes |
|---|---|---|---|---|---|---|
| Model | 1.0 | 1.077 | 1.077 | 132.506 | <0.0001 | *** |
| Error | 115.0 | 0.935 | 0.008 | | | |
| Corrected Total | 116.0 | 2.011 | | | | |

Notes: Computed against model Y = Mean (Y). Signification codes: 0 < *** < 0.001.

In other words, the groundwater level is not the same everywhere. It varies depending on the location. The ANOVA table shows that the location of the groundwater level measurements explains 1.077 m of the variation in the data. This is a significant amount of variation, and it means that the location of the measurements is a factor that can affect the groundwater level.

Table 8 below is a table of model parameters for groundwater level. Table 8 has six columns:

- Source—This column identifies the source of the parameter. In this case, the source of the parameters is the model;
- Value—This column shows the value of the parameter. In this case, the value of the intercept is 1.464 and the value of soil surface temperature (SST), °C is 0.019;
- Standard error—This column shows the standard error of the parameter. In this case, the standard error of the intercept is 0.052 and the standard error of SST, °C is 0.002;
- t—This column is the t-statistic for the parameter. The t-statistic is a measure of the significance of the parameter. In this case, the t-statistic for the intercept is 28.324 and the t-statistic for SST, °C is 11.511;
- Pr > |t|—This column is the *p*-value for the parameter. The *p*-value is a measure of the probability of obtaining the observed results if the null hypothesis is true. In this case, the *p*-value for the intercept is <0.0001 and the *p*-value for SST, °C is <0.0001;
- Signification codes—This column shows the significance of the results. A significance code of * indicates that the results are significant at the 0.05 level. A significance code of ** indicates that the results are significant at the 0.01 level, and a significance code of *** indicates that the results are significant at the 0.001 level.

In this case, the *p*-values for both the intercept and SST, °C are less than 0.05. This means that the results are significant at the 0.05 level. Therefore, we can conclude that both the intercept and SST, °C are significant factors that can affect the groundwater level.

**Table 8.** Model parameters (groundwater level, m).

| Source | Value | Standard Error | t | Pr > \|t\| | Lower Bound (95%) | Upper Bound (95%) | *p*-Values Signification Codes |
|---|---|---|---|---|---|---|---|
| Intercept | 1.464 | 0.052 | 28.324 | <0.0001 | 1.362 | 1.566 | *** |
| Soil surface temperature (SST), °C | 0.019 | 0.002 | 11.511 | <0.0001 | 0.016 | 0.023 | *** |

Note: Signification codes: 0 < *** < 0.001.

In other words, the groundwater level is not the same everywhere. It varies depending on the location and the temperature. Table 8 shows that the intercept is 1.464, which means that the groundwater level is expected to be 1.464 m above sea level if the temperature is 0 °C. Table 8 also shows that SST, °C is 0.019, which means that the groundwater level is expected to increase by 0.019 m for every 1 degree Celsius increase in temperature.

It is important to note that these are just the estimated values of the parameters. The actual values of the parameters may be different.

Table 9 shows the results of a regression analysis of the variable groundwater level. The analysis was conducted on 117 observations, with a sum of weights of 117. The degrees of freedom (DF) are 115. The $R^2$ value is 0.535, which indicates that 53.5% of the variation in the groundwater level variable is explained by the regression model (Figure 3). The adjusted $R^2$ value is 0.531, which is a more accurate measure of the model's fit, as it takes into account the number of variables in the model. The mean squared error (MSE) is 0.008, which indicates that the average error between the predicted and actual values of the groundwater level variable is 0.008. The root mean squared error (RMSE) is 0.090, which is a more interpretable measure of the error, as it is in the same units as the groundwater level variable. The mean absolute percentage error (MAPE) is 3.581, which indicates that the average absolute percentage error between the predicted and actual values of the groundwater level variable is 3.581%. The Durbin–Watson (DW) statistic is 1.529, which indicates that there is no evidence of autocorrelation in the residuals. The Cp statistic is 2.000, which indicates that the model is not overfit. The Akaike information criterion (AIC) is −561.094, which indicates that the model has a good fit to the data. The Schwarz Bayesian criterion (SBC) is −555.570, which indicates that the model has a good fit to the data. The partial correlation coefficient (PC) is 0.481, which indicates that the groundwater level variable is moderately correlated with the other variables in the model.

**Table 9.** Regression of variable groundwater level.

| Goodness of Fit Statistics (Groundwater Level, m) | |
|---|---|
| Observations | 117 |
| Sum of weights | 117 |
| DF (Degrees of freedom) | 115 |
| $R^2$ | 0.535 |
| Adjusted $R^2$ | 0.531 |
| MSE (Mean squared error) | 0.008 |
| RMSE (Root mean squared error) | 0.090 |
| MAPE (Mean absolute percentage error) | 3.581 |
| DW (Durbin–Watson) | 1.529 |
| Cp (Mallows's Cp coefficient) | 2.000 |
| AIC (Akaike information criterion) | −561.094 |
| SBC (Schwarz Bayesian criterion) | −555.570 |
| PC (Partial correlation coefficient) | 0.481 |

In summary, the regression analysis of the variable groundwater level indicates that the model is a good fit to the data and that the groundwater level variable is moderately correlated with the other variables in the model.

As can be seen from this Figure 6, as the temperature of the soil surface increases, the groundwater level decreases.

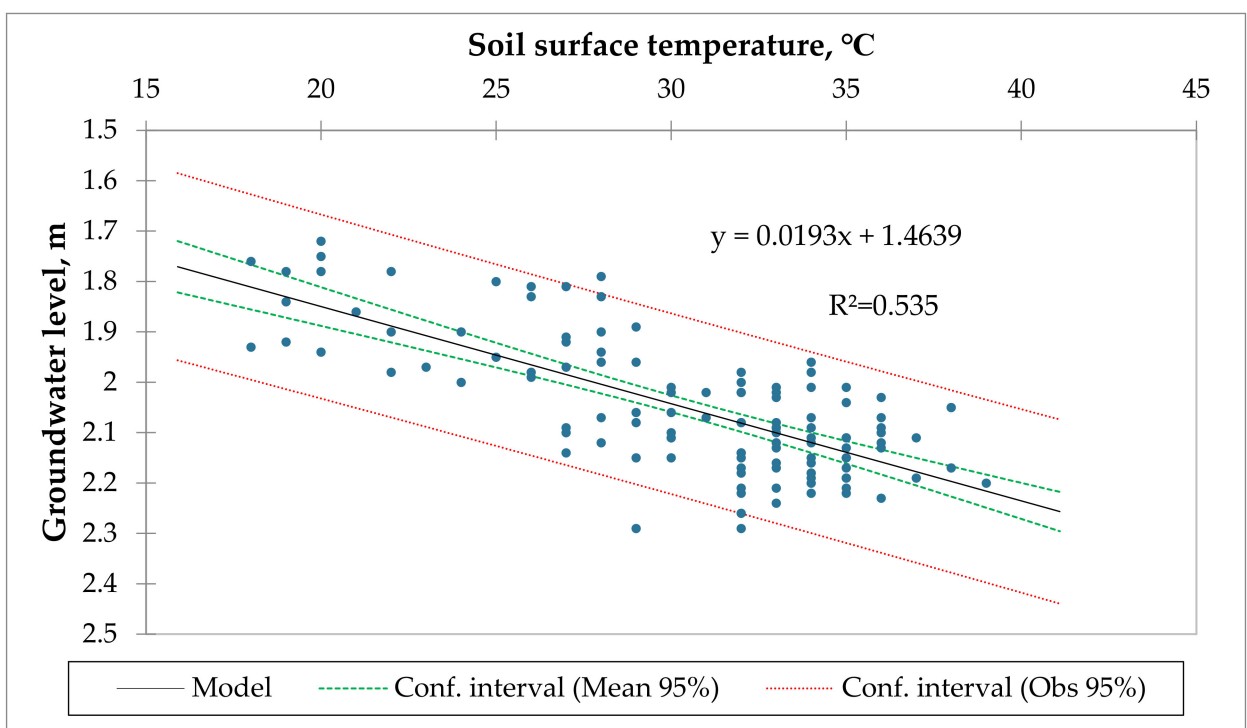

**Figure 6.** The relationship between soil surface temperature and groundwater level in the Bukhara region of Uzbekistan.

The article discusses the relationship between soil surface temperature and groundwater level, and its impacts on ecosystems and human society. The results show that soil surface temperature and groundwater levels varied significantly over time and were influenced by various factors such as the type of soil, amount of vegetation, water in the soil, wind speed, amount of sunlight, and air temperature. Analysis elsewhere has also indicated that soil surface temperature can be a good proxy for assessing the groundwater level, especially in tropical peatlands [45].

Statistical analysis using a linear regression model and ANOVA tests showed that both soil surface temperature and the location of groundwater level measurements were significant factors that affect the groundwater level. The correlation between the soil surface temperature and groundwater level was found to have an $R^2$ value of 0.428 in other research [24]. Moreover, the correlation between the soil temperature and groundwater level was $R^2 = 0.49$ when the soil temperature was measured at 5 cm, and $R^2 = 0.59$ when the soil temperature was measured at 10 cm [46].

The analysis also showed that 53.5% of the variation in the groundwater level variable is explained by the regression model, indicating a moderately correlated relationship between the groundwater level and soil surface temperature in the model. Additionally, the ANOVA regression results from Indian Punjab suggest that groundwater level may also vary significantly with respect to agro-climatic regions, crop diversity, and farmer education [47].

The impacts of the relationship between the soil surface temperature and groundwater level include reduced water availability, increased risk of waterborne diseases, reduced habitat for groundwater-dependent plants and animals, and increased erosion of shorelines

and stream banks. The findings of this study are important in developing an understanding of the effects of climate change on these important environmental variables and can be used to develop sustainable water management strategies. Climate change has also caused the average temperature of the earth's atmosphere to increase, and it has changed the way that precipitation falls around the world. These changes can directly affect the levels of groundwater [48]. Machine learning models can also provide valuable insights into the prediction of groundwater levels, especially in water-deficient areas [49], like in Uzbekistan. Finally, the findings of this research are in line with the hypothesis set in the beginning of this manuscript: soil surface temperature has a significant impact on changes in groundwater level.

## 4. Conclusions

The findings of this study demonstrate the importance of considering the soil surface temperature when managing groundwater resources. The reduction in groundwater level due to the increase in soil surface temperature can have significant negative impacts on ecosystems and human societies. The use of linear regression analysis and ANOVA tests provides a statistical basis for understanding and managing the relationship between groundwater level and soil surface temperature. Through the long-term monitoring and management of groundwater resources, it is possible to mitigate the negative impacts of changing soil surface temperatures and maintain the availability and quality of groundwater resources for future generations. The general trend is that as the temperature of the soil surface increases, the level of groundwater decreases. However, there are many exceptions to this trend, and it is important to consider all of the factors that may be involved in any given situation.

The findings further highlight that the relationship between soil surface temperature, groundwater level, and solar radiation is complex and depends on a number of factors, including soil type, vegetation cover, and climate patterns. In general, higher solar radiation leads to higher soil surface temperature and higher evapotranspiration rates, which can lead to a decrease in groundwater level. As a result, we see that the soil surface temperature determines changes in groundwater level in the study region.

Based on the findings of the study, we recommend the following actions be implemented:

- By monitoring groundwater levels and soil surface temperatures, it is possible to identify areas where groundwater resources are at risk of depletion. This information can then be used to warn stakeholders and develop mitigation strategies. Farmers could use sensors to monitor soil surface temperature and groundwater levels in their fields. This information can then be used to adjust irrigation practices and other land management practices to conserve groundwater resources;
- Practices such as crop rotation, cover cropping, and reduced tillage can help to reduce soil surface temperature and evaporation rates, which can help to conserve groundwater resources;
- Water managers could develop a network of monitoring stations to track groundwater levels and soil surface temperatures across a region. These data could then be used to create early warning systems for groundwater depletion and to develop more effective management plans;
- The government could provide incentives to farmers and other land users to adopt sustainable land management practices that conserve groundwater resources.

**Author Contributions:** M.K.: designed the research concept and methodology, carried out the analysis, interpreted the results, and prepared the original draft. J.I.: designed the research concept and methodology, carried out the analysis, interpreted the results, and prepared the original draft. A.H.: supported the analysis, interpreted the results, and reviewed and edited during the writing stage. E.S.: designed the research concept and methodology, interpreted the results, and supported in addressing the reviewers' comments. Z.G.: supported in addressing the reviewers' comments. All authors have read and agreed to the published version of the manuscript.

**Funding:** This research study was funded by the Ministry of Higher Education, Science and Innovations of the Republic of Uzbekistan (FZ-20200929192).

**Data Availability Statement:** The data that support the findings of this study can be requested from the corresponding author.

**Acknowledgments:** This research was implemented in the frame of "Development of methods for assessing and forecasting the impact of global climate change on the reclamation of irrigated lands for the efficient use of water resources and the creation of electronic maps of hydromodular zoning of irrigated lands based on GIS technology" project. Ahmad Hamidov received funding from the German Ministry of Education and Research (BMBF) within the framework of the SusWEF (Sustainable water-saving irrigation technologies for achieving water, energy, and food security in the context of climate change in Uzbekistan (FKZ: 01DK22002)). This publication was supported under the CGIAR Initiative on NEXUS Gains, which received the support of CGIAR Trust Fund contributors.

**Conflicts of Interest:** The authors declare no conflict of interest.

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
