# Peer review of "Impact of Soil Surface Temperature on Changes in the Groundwater Level"

_water, doi:10.3390/w15213865_

Round 1

Reviewer 1 Report (Previous Reviewer 1)

Comments and Suggestions for Authors

No more comments.

Author Response

The authors thank the reviewers for time and the effort spent to improve the manuscript. The efforts are greatly acknowledged. We agree in general by all points raised by the reviewer and have revised the manuscript accordingly. Please see attached the document for our detailed responses.

Reviewer 2 Report (Previous Reviewer 3)

Comments and Suggestions for Authors

The authors improved the manuscript according to the previous comments and suggestions i proposed. They should conduct minors revisions by including the comments and suggestions of the Editors and other reviewers as well.

Author Response

The authors thank the reviewers for time and the effort spent to improve the manuscript. The efforts are greatly acknowledged. We agree in general by all points raised by the reviewer and have revised the manuscript accordingly. Please see attached the document for our detailed responses.

Reviewer 3 Report (New Reviewer)

Comments and Suggestions for Authors

1) Lines 37-42: How soil moisture and soil temperature are related? The explanation is needed. You can mention that “Reducing soil moisture and increasing soil temperature at the soil surface can be attributed to the removal of canopy cover and loss of organic matter.” Please see this paper [DOI 10.3389/fenvs.2023.1213181]

2) Lines 95-97: These sentences can be moved to “2. Materials and Methods”.

3) Line 120: Delete “(own graph)”.

4) Table 1: What is mean for “#”? Please revise.

5) Section “2.2. Linear Regression model”: Why linear regression was used in this study? Please explain more details.

6) Figure 2: It needs to show the linear equation, which can present the rate of change.

 7) Lines 426-431: These sentences need the support from the previous. Please see this paper [https://doi.org/10.3390/w15193473]

8) Abstract should be provided more details of the key finding.

9) “4. Conclusions”: Could you recommend for the reality implementation based on your finding?

Comments on the Quality of English Language

-

Author Response

The authors thank the reviewers for time and the effort spent to improve the manuscript. The efforts are greatly acknowledged. We agree in general by all points raised by the reviewer and have revised the manuscript accordingly. Please see attached the document for our detailed responses.

Reviewer 4 Report (New Reviewer)

Comments and Suggestions for Authors

GENERAL COMMENTS

The work entitled “Impact of Soil Surface Temperature on the Changes of Groundwater Level”

RELEVANCE (considering the contribution to the advancement of knowledge): Good

ORIGINALITY (considering the problem to be studied and the existing knowledge gaps that justify the study): Good

TECHNICAL AND SCIENTIFIC MERIT: Good

FINAL OPINION:

The work has potential and merit.

Introduction

The Introduction is good and needs no adjustments.

What are the hypotheses of the work?

Material and Methods

This item needs adjustments, review.

What type of soils?

What is the soil classification system?

Results

The work presents the data adequately.

Discussion

The authors discussed the results well.

Conclusions

The conclusions are consistent with the results.

References

References are compatible with the Work.

Comments on the Quality of English Language

OK

Author Response

The authors thank the reviewer for time and the effort spent to improve the manuscript. The efforts are greatly acknowledged. We agree in general by all points raised by the reviewer and have revised the manuscript accordingly. Please see attached the document for our detailed responses.

Round 2

Reviewer 3 Report (New Reviewer)

Comments and Suggestions for Authors

Accept in present form.

Comments on the Quality of English Language

-

This manuscript is a resubmission of an earlier submission. The following is a list of the peer review reports and author responses from that submission.

Round 1

Reviewer 1 Report

Comments and Suggestions for Authors

1) Authors should explained location of monitoring points (117???), exact time of monitoring (period in year), air temperature, soil characteristics, and other climatological parameters.

2) Long lasting time series for the 1991-2020 period of air temperature, precipitation and groundwater level during the growing season as well as during the whole year should be analysed.

Reviewer 2 Report

Comments and Suggestions for Authors

Hello,

In the conclusions section, the authors wrote: "The findings of this study demonstrate the importance of considering soil surface temperature when managing groundwater resources. …". I agree. The issue is that it is worth an extended study (e.g., about soil type information in the measurement points, three axes (three factors) analysis on the one figure would be interesting and helpful in the interpretation). The analyzed problem is vital cause it gives the possibility to monitor a huge amount of area (even in the automatic approach based on satellite data). Therefore, the application can be significant, e.g., flood embankments.

Some editorial mistakes (like in the affiliation, tables 2 and 7, no abbreviation explanations). Some methodological aspects need to be revised (e.g., there is no need to explain the groundwater level in the Water journal, which includes hydrogeology. The same is about the linear regression model). I am not sure that the country's history must be mentioned in the scientific paper (108 and 109 lines).  

Eventually, the issue is whether soil surface temperature determines changes in groundwater level or whether groundwater level determines changes in soil surface temperature. Please analyze it in reference to solar radiation. Give some remarks about this in the paper.

Best regards

Comments on the Quality of English Language

As written above, moderate editing of the English language is required. However, redaction analyses need to be done carefully.

Reviewer 3 Report

Comments and Suggestions for Authors

The manuscript “Impact of Soil Surface Temperature on The Changes of Groundwater Level” submitted by the authors analysis the relationship between the variation of surface temperature and groundwater level. The subject is of the greatest interest and if accepted could provide a huge contribution to researchers dealing with the impact of climate changes. However, the current study is not well written and well presented. The authors should provide additional details before it could be accepted for publication. Below are some comments and suggestions to improve the overall quality of this manuscript.

Lines 12-22: The abstract is not well written and well organized. Because here, the authors presented the findings begore they could present the methodology. The authors should first provide brief introduction of the study, its novelty, the applied methodology, their findings, and results and after the main conclusions.  

On lines 84-85, authors state that the aim of the current research study is to examine the correlation between soil surface temperature and groundwater level through conducting on-site observations, but in the section of Material and methods the authors don’t provide details and descriptions on these datasets. They should provide such comprehensive details and descriptions. The authors only provide information on linear regression model.

Comments on the Quality of English Language

Moderate editing of English language are required